# Motivating Structured walking Activity in people with Intermittent Claudication (MOSAIC): protocol for a randomised controlled trial of a physiotherapist-led, behavioural change intervention versus usual care in adults with intermittent claudication

Lindsay Bearne,[1] Melissa Galea Holmes,[1,2] Julie Bieles,[1] Saskia Eddy,[1] Graham Fisher,[1] Bijan Modarai,[3] Sanjay Patel,[4] Janet L Peacock,[1] Catherine Sackley,[1] Brittannia Volkmer,[1] John Weinman[5]

For numbered affiliations see end of article.

**Correspondence to**
Dr Lindsay Bearne;
lindsay.bearne@kcl.ac.uk

## ABSTRACT

**Introduction** Walking exercise is a recommended but underused treatment for intermittent claudication caused by peripheral arterial disease (PAD). Addressing the factors that influence walking exercise may increase patient uptake of and adherence to recommended walking. The primary aim of this randomised controlled trial (RCT) is to evaluate the efficacy of a physiotherapist-led behavioural change intervention on walking ability in adults with intermittent claudication (MOtivating Structured walking Activity in people with Intermittent Claudication (MOSAIC)) in comparison with usual care.

**Methods and analysis** The MOSAIC trial is a two-arm, parallel-group, single-blind RCT. 192 adults will be recruited from six National Health Service Hospital Trusts. Inclusion criteria are: aged ≥50 years, PAD (Ankle Brachial Pressure Index ≤0.90, radiographic evidence or clinician report) and intermittent claudication (San Diego Claudication Questionnaire), being able and willing to participate and provide informed consent. The primary outcome is walking ability (6 min walking distance) at 3 months. Outcomes will be obtained at baseline, 3 and 6 months by an assessor blind to group allocation. Participants will be individually randomised (n=96/group, stratified by centre) to receive either MOSAIC or usual care by an independent randomisation service. Estimates of treatment effects will use an intention-to-treat framework implemented using multiple regression adjusted for baseline values and centre.

**Ethics and dissemination** This trial has full ethical approval (London—Bloomsbury Research Ethics Committee (17/LO/0568)). It will be disseminated via patient forums, peer-reviewed publications and conference presentations.

**Trial registration number** ISRCTN14501418

## Strengths and limitations of this study

► This is the first trial to investigate the efficacy of a physiotherapist-led behavioural change intervention on walking ability in people with intermittent claudication.
► This trial collects validated objective and self-reported outcomes of walking ability.
► Clinician's experiences of training and delivering MOtivating Structured walking Activity in people with Intermittent Claudication and intervention fidelity will be explored to inform implementation into practice.
► This trial only follows participants for 6 months post-randomisation.

## INTRODUCTION
### Background and objectives

Walking exercise is an effective treatment for intermittent claudication,[1] an ischaemic leg pain caused by peripheral arterial disease (PAD). It improves walking distances and duration compared with usual National Health Service (NHS) treatment[2] or pharmaceutical therapy[3] and has comparable long-term outcomes to revascularisation.[4 5] The Trans-Atlantic Inter-Society Consensus-II group recommend supervised walking exercise at an intensity that induces pain within 3–5 min, for 30–60 min/session conducted three times/week for 3 months.[6] Similarly, the National Institute for Health and Care Excellence recommends 2 hours of supervised exercise/week for 3 months.[7] However, guideline implementation is poor[8–10] and, even when

supervised exercise therapy is available, patient uptake and adherence is variable.[11] Consequently, the usual care for most individuals with intermittent claudication is simple walking advice, however, despite advice, self-directed walking is frequently overlooked as a management strategy,[12] participation is low[10 13] and often ineffective.[14]

Adopting and maintaining a new health behaviour, such as walking exercise, is challenging and people with intermittent claudication frequently do not achieve walking recommendations.[11 15 16] Barriers to walking exercise in people with intermittent claudication include the need to stop and rest to alleviate pain, the impact of undulating terrain and a lack of clarity around the recommendations for walking.[13 17] These challenges can be difficult to overcome without appropriate guidance and support.

Structured home-based exercise programmes may offer a promising alternative to supervised exercise therapy.[18 19] These programmes typically include exercise sessions completed away from a healthcare facility with support from healthcare professionals and may overcome the barriers of accessibility and availability.[14 18 20] Evidence from recent systematic reviews suggests that structured home exercise may improve walking performance although the quality of the included trials is mixed and few studies evaluate systematically developed interventions which incorporate theoretically informed strategies to facilitate the uptake and long-term adherence to walking exercise.[14 20]

Crucial conditions to support health behavioural change, such as increasing self-directed walking exercise, are an individual's capability (eg, knowledge and understanding that walking exercise is a treatment for intermittent claudication), opportunity (eg, identifying an appropriate environment place to walk) and motivation (eg, attitudes and beliefs about walking as a treatment).[21–23] Evidence suggests that targeting these factors using behavioural change techniques (eg, setting walking goals, action planning[22 24] or motivational interviewing[25]) in addition to exercise or advice may increase walking ability in people with intermittent claudication.[24 26]

Motivating Structured walking Activity in people with Intermittent Claudication (MOSAIC) is a structured, physiotherapist-led behavioural change intervention, which aims to increase walking in people with intermittent claudication. It is informed by two psychological models (1) the theory of planned behaviour[27 28] and (2) the common-sense model of illness representations.[29 30] The theory of planned behaviour proposes that people use information (eg, social, personal and environmental conditions) around them to make decisions about whether to perform a behaviour (eg, walking exercise).[27 28] Interventions underpinned by the theory of planned behaviour have shown utility in physical activity behavioural change,[31] and the sociocognitive factors defined by this model are associated with motivation to walk and walking ability in adults with intermittent claudication.[23 32] The common-sense model of illness representations proposes that individuals form personal, lay explanations of their illness (eg, PAD) and symptoms (eg, intermittent claudication), which guides how they manage their condition and seek treatment.[29 30] The common-sense model has been instrumental in understanding how people cope with long-term illness.[33] Illness perceptions defined by the common-sense model, including beliefs about PAD, its causes and one's ability to control or manage the condition and symptoms, are associated with walking ability in people with intermittent claudication.[23]

MOSAIC was systematically developed to target the salient factors identified from these two models[12 23] and includes behavioural change techniques to facilitate the uptake and maintenance of walking exercise.[24] This protocol was informed by a preliminary study which demonstrated that the MOSAIC intervention and trial procedures were acceptable to participants and clinicians and feasible to deliver.[26]

## Objectives

The primary objective is to answer the question: Does MOSAIC improve walking ability (measured by the 6 min walking distance, 6MWD) at the primary endpoint of 3 months compared with usual care in older adults with intermittent claudication?

The secondary objectives include answering the questions:

1. Does MOSAIC improve (a) activities of daily living and quality of life (QoL) at 3 months and (b) walking ability, activities of daily living and QoL at 6 months compared with usual care in older adults with intermittent claudication?
2. Is it feasible to collect resource use data using a modified measure in older adults with intermittent claudication?
3. What are the minimal clinically important difference (MCID) values for the clinical assessments used for older adults with intermittent claudication?

## METHODS

### Trial design

A phase II, prospective, assessor blinded, multicentre, parallel group, two-arm, randomised, controlled superiority trial with nested qualitative study (figure 1).

### Participant eligibility criteria

Individuals will be eligible for the trial if they meet the following inclusion criteria: (1) adults ≥50 years of age; (2) established PAD (determined by either Ankle Brachial Pressure Index ≤0.90, radiographic evidence or clinician report) and intermittent claudication (reported on the San Diego Claudication Questionnaire[34]); (3) able and willing to participate in MOSAIC and provide informed consent.

Individuals may not enter the study if they meet the following exclusion criteria: (1) unstable intermittent claudication (eg, self-reported change in symptoms during previous 3 months in response to the question 'have your symptoms changed in the last 3 months?';

(2) walking >90 min/week (reported on the Brief International Physical Activity Questionnaire (IPAQ)[35]); (3) contraindications to walking exercise (eg, unstable angina) confirmed by the direct care team; (4) have completed any prescribed supervised exercise sessions in the previous 6 months or been offered prescribed supervised exercise sessions in the next 6 months.

## Setting

Participants will be recruited from five NHS Foundation Trusts in London (Guy's and St Thomas', King's College, St George's, Royal Free, Royal London) and one in the South East of England (Ashford and St Peter's), UK. The MOSAIC intervention is delivered either at the participants' home or a private room in the participating site to allow flexibility, inclusivity and confidentiality, while minimising contamination.

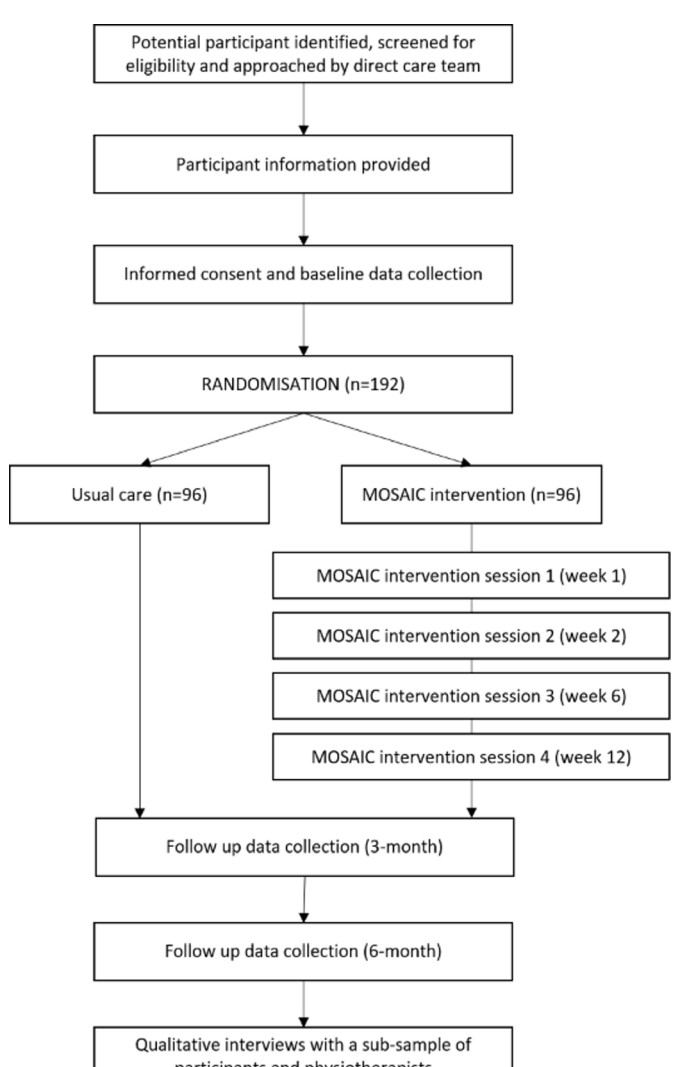

**Figure 1** Trial flow chart.

## Interventions

### MOtivating Structured walking Activity in people with Intermittent Claudication

Participants randomised to the intervention group will receive the MOSAIC intervention in addition to the usual care provided by their direct care team.

MOSAIC aims to increase an individual's walking ability. It comprises two 60 min face-to-face consultations (weeks 1 and 2) and two 20 min follow-up telephone calls (weeks 6 and 12). The content of each session is standardised and incorporates 15 behavioural change techniques, but sessions may be tailored based on participants' knowledge, goals, symptoms and current walking (tables 1 and 2). Walking plans will be agreed collaboratively between the participant and the physiotherapist and include progressive, individualised targets for walking frequency, intensity and duration to achieve recommendations (30–50 min of walking three times/week at an intensity that elicits pain within 3–5 min).[6] MOSAIC is supplemented by an interactive participant manual, containing worksheets and a walking diary, and by a pedometer.

### Physiotherapist training and supervision

The MOSAIC sessions will be delivered by experienced physiotherapists (band 6 and above) who will be trained in motivational interviewing and behavioural change techniques. All trial physiotherapists will receive 2 days training on the trial objectives, research processes, underpinning psychological theories and MOSAIC intervention content, delivery and materials. Trial physiotherapists will be explicitly instructed on how to identify and report any adverse events and about the risks and consequences of contamination. Training will be delivered by the trial team (LB, JW, MGH, JB, GF and BV) and an accredited provider of motivational interviewing training (Pip Mason Consultancy). A bespoke manual will support training and delivery of MOSAIC and includes an intervention checklist for each session.

To promote consistency of intervention delivery, all face-to-face and telephone sessions will be audiorecorded and individualised feedback on at least one of the first face-to-face and telephone MOSAIC sessions will be provided to each physiotherapist by a member of the trial team (LB and BV). Physiotherapists will record each session on a checklist (eg, duration, content), which will be reviewed by members of the trial team for adherence to the intervention. The trial physiotherapists will attend regular meetings supervised by members of the trial team (JW, LB and BV) to maintain skills and receive feedback and support.

### Usual care

Participants randomised to the comparison group will continue to receive usual care provided by the NHS for intermittent claudication. Usual care is typically delivered in the vascular outpatient clinic by a vascular surgeon or specialist nurse and may include: the provision of

**Table 1** Format and content of the MOtivating Structured walking Activity in people with Intermittent Claudication (MOSAIC) intervention

**1. Session 1 (60 min face to face)**

Establish mutual objectives for session. Identify participant expectations, reframe unrealistic expectations, summarise format and how MOSAIC might be helpful.

Elicit participant experiences and understanding of their disease/symptoms. With permission, provide relevant information.

Elicit treatment experiences. Discuss any treatment the participant has attempted and introduce walking as an option.

With permission, provide further information about walking as treatment and elicit the participant's thoughts on this information.

With permission, provide standard walking recommendations for intermittent claudication. Invite participant thoughts on this new information, provide reassurance, if needed.

Introduce participant manual, highlighting the key information a participant may wish to review and consider before next session.

**2. Session 2 (60 min face to face)**

Establish mutual objectives for session. Review session 1 and confirm/ reframe participant expectations, as required.

Evoke discussion about participant's current walking, reinforcing change talk by affirming positive efforts and successes.

Elicit or reinforce change talk, supporting increased readiness to walk and addressing any signs of uncertainty.

Shift focus to mobilisation and commitment to change.

Introduce goal setting and discuss role of walking exercise in achieving valued activities (eg, hobbies); agree a goal to increase walking.

Introduce action planning and agree and record a tailored exercise prescription and walking action plan in the participant manual.

Address problem solving. Identify potential barriers to goals and measures for overcoming anticipated or unexpected obstacles using participant manual.

Introduce self-monitoring and provide walking self-monitoring tools. Elicit participant interest in and confidence to use these tools.

**3. Session 3 (20 min telephone call)**

Review of previous sessions and establish mutual objectives for telephone call.

Review and provide feedback on progress toward goals using the goal setting sheet, affirm positive attempts and efforts.

Elicit discussion and reflection on self-monitoring skills. Affirm efforts to self-monitor.

Problem solve new issues by eliciting reflection on anticipated and actual barriers and problem solve collaboratively.

Review and accept or revise walking goals and action plans, as necessary.

Support maintenance. discuss and define relapse. Discuss any new barriers and measures to overcome these and identify strategies to re-engage.

Generalise skills and behaviour. discuss ways that walking can be incorporated to daily life. Elicit discussion on how skills can be applied to new health goals.

**4. Session 4 (20 min telephone call)**

Review of previous sessions and establish mutual objectives for telephone call.

Review and provide feedback on progress toward goals using the goal setting sheet, affirm positive attempts and efforts.

Elicit discussion and reflection on self-monitoring skills. Affirm efforts to self-monitor.

Problem solve new issues by eliciting reflection on anticipated and actual barriers and problem solve collaboratively.

Review and accept or revise walking goals, as necessary.

Discuss and revise goals and action plan, as necessary.

Support maintenance. Discuss any new barriers and measures to overcome these and identify strategies to re-engage.

Generalise skills and behaviour. discuss how walking can be incorporated to daily life.

Elicit discussion on how skills can be applied to new health goals. Signpost to community resources.

**Table 2** Behavioural change techniques included in MOtivating Structured walking Activity in people with Intermittent Claudication intervention

| Behavioural change technique | Definition | Delivery of behavioural change technique | Example of mapped construct from the TPB or CSM |
|---|---|---|---|
| Social support (general) | Advise on, arrange or provide social support, or non-contingent praise or reward for performance of the behaviour, and encouragement and counselling when directed at the behaviour (eg, motivational interviewing) | Physiotherapist trained in motivational interviewing, values-based goals elicited to support autonomy, change talk facilitated through patient-centred dialogue | Subjective norm (TPB) and the personal schematic illness representation (CSM) |
| Information about health consequences | Provide information about health consequences of performing the behaviour | Potential benefits of walking discussed, including ability to walk further before pain onset or need to stop and rest. | Attitude (TPB), treatment control (CSM) |
| Reduce negative emotions | Advise on ways of reducing negative emotions to facilitate performance of the behaviour | Elicit and address worry or fear about walking, assuring participant that IC is not a sign of damage to the limb, if required | Emotional response (CSM), attitude (TPB) |
| Framing/reframing | Suggest the deliberate adoption of a perspective or new perspective on behaviour (eg, its purpose) to change cognitions or emotions about performing the behaviour | If the participant reports a belief about walking, which is not consistent with evidence review the topic and suggest alternatives to the participant to support their understanding | Attitude (TPB) and coherence (CSM) |
| Focus on past success | Advise to think about or list previous successes in performing or attempting the behaviour | Highlight and reinforce successful attempts at walking (or other health behaviour if no walking attempted), even if goals are not fully achieved | Perceived behavioural control (TPB) |
| Goal setting (behaviour) | Set or agree on a goal defined in terms of the behaviour to be achieved | Walking goal defined in terms of frequency, duration, intensity and context. Based on current walking with the aim of progressing toward recommended walking level | Intention (TPB) |
| Goal setting (outcome) | Set or agree on a goal defined in terms of a positive outcome of wanted behaviour | Value-based goal identified which would be facilitated by improved walking (eg, work, hobby, social activity) | Intention (TPB) |
| Instruction on how to perform behaviour | Advise or agree on how to perform the behaviour (walking) | Discuss the walking recommendations with the participant | Treatment control (CSM), attitude (TPB) |
| Problem solving | Analyse or prompt analysis of factors influencing the behaviour and generate or select strategies that include overcoming barriers and / or increasing facilitators | Participants encouraged to identify barriers which may prevent them achieving their goal, and realistic solutions discussed and agreed | Perceived behavioural control (TPB) |
| Action planning | Prompt detailed planning of behaviour performance, including at least one of: context, frequency, duration or intensity | Action plan worksheet completed, recording details of the context, frequency, duration and intensity of walking goal | Intention–behaviour translation (TPB) |

Continued

**Table 2** Continued

| Behavioural change technique | Definition | Delivery of behavioural change technique | Example of mapped construct from the TPB or CSM |
|---|---|---|---|
| Self-monitoring of behaviour | Establish a method for the person to monitor or record their behaviour as part of a behavioural change strategy | Physiotherapist discussed methods to monitor daily walking (eg, wearing a watch, or using landmarks to note distance achieved) | Self-regulation (CSM) and intention-behaviour translation (TPB) |
| Review behavioural goals | Review behavioural goals jointly and consider modifying goal or behavioural change strategy in light of achievement | Walking discussed relative to goals and revised as appropriate to be more achievable or challenging | Self-regulation (CSM) and intention-behaviour translation (TPB) |
| Review outcome goals | Review outcome goals jointly and consider modifying goals in light of achievement | Value-based goal considered relative to walking and revised if no longer salient | Self-regulation (CSM) and intention-behaviour translation (TPB) |
| Feedback on behaviour | Monitor and provide informative or evaluative feedback on performance of the behaviour | Walking completed by participant is discussed considering individual goals and recommended walking treatment for intermittent claudication | Self-regulation (CSM) and intention-behaviour translation (TPB) |
| Generalisation of a target behaviour | Advise to perform the wanted behaviour, which is already performed in a particular situation, in another situation | Together, identify ways that walking can be incorporated to daily life in a way that will maintain or improve intermittent claudication. | Personal control (CSM), perceived behavioural control |

CSM, common sense model; IC, intermittent claudication; TPB, theory of planned behaviour.

information about PAD and lifestyle modifications (eg, smoking cessation, diet and weight management), supervised exercise, risk factor management including lipid modification, statin therapy and antiplatelet therapy, pharmacotherapy to improve leg function (vasodilators such as naftidrofuryl oxalate) or revascularisation (eg, angioplasty, stenting and bypass).[7] Participants may seek concomitant treatment during the trial if they wish to. Any other treatments accessed by participants, (eg, from their general practitioner/other health professionals), will be recorded on a modified Client Service Receipt Inventory[36] at follow-up.

### Participant identification and recruitment
Potentially eligible patients will be identified by one of two methods which will proceed in parallel:
1. Patients will be approached by the direct care team during routine clinical appointments at participating sites. Potential participants will be provided with an explanation of the trial, an invitation letter and participant information sheet and, if interested, asked to provide permission to be contacted by the study recruitment personnel.
2. Patients will be identified from existing clinical lists/databases (depending on availability of these at participating sites) and will be sent an invitation pack including: an invitation letter, participant information sheet, consent to be contacted form and prepaid return envelope. Non-responders will be contacted by telephone approximately 4 weeks later.

Patients expressing an interest in participating in the trial will be contacted by the study recruitment personnel to complete full eligibility screening. Eligible patients who do not wish to take part in the trial will be asked if their age and gender may be recorded and if they wish to provide a reason for opting not to participate.

Eligible patients will be invited to attend an appointment at their participating site to provide written informed consent and complete a baseline assessment.

### Randomisation, blinding and allocation concealment
Outcome measures will be collected by an independent assessor who will attend each site and will be masked to group allocation. The independent assessor will not be involved in delivering MOSAIC.

Following baseline data collection, the independent assessor will notify a web-based randomisation service to randomise each participant. The randomisation service is provided by the UK Clinical Research Collaboration Registered King's Clinical Trials Unit and will ensure prospective registration and allocation concealment. Randomisation will be at the level of the individual, using block randomisation with randomly varying block sizes, stratified by centre. The online randomisation system will generate emails automatically to delegated members of the research team and the allocating trial physiotherapists at the study sites. The allocating trial physiotherapists will coordinate the delivery of MOSAIC at the study site but will not share this information with the independent assessor.

The recruiting personnel and independent assessor are masked to group allocation. The trial statistician, the research staff undertaking the qualitative study, the participants and the trial physiotherapists are not masked to group allocation.

### Outcome assessment
Measures will be conducted immediately before randomisation (baseline), and at 3 months and 6 months post-randomisation (table 3).

### Data collection
Written informed consent will be obtained prior to any data collection. Baseline and 3-month follow-up measures will be conducted face to face by an independent assessor who will attend each site. At 6 months, the outcomes will be collected either electronically or via a standard postal questionnaire pack (with prepaid return envelope).

Data will be collected and stored using a secure database system. Participants opting to complete self-reported assessments electronically will be provided with a unique username and password to log onto the database and complete measures either during the face-to-face appointment (baseline and 3 months) or from home (at 6 months). Any research data completed by pencil and paper will be entered to the secure database by the independent outcome assessor or trial research assistant.

At 3-month follow-up, participants who are unable or refuse to attend an appointment to complete the primary outcome will be invited to complete the secondary outcomes at home either electronically or by post.

Attrition will be minimised via standardised text, email and telephone reminders. Following these, if no data have been returned, the assessor will telephone the participant to collect a minimum data set (table 3).

### Baseline measures
At baseline, participants will complete a bespoke sociodemographic and clinical characteristics questionnaire (including age, sex, smoking history, comorbidities and claudication symptom classification (San Diego Claudication Questionnaire[34]). Standard measures of body mass index and Ankle-Brachial Pressure Index[37] will also be determined in addition to the outcomes detailed below.

### Primary outcome
The primary outcome is the difference in mean 6MWD in metres at 3 months between groups. This will be assessed by a self-paced 6 min walk test over a 60.96 m (200 foot) circuit.[38–40] Pain intensity will be measured before and after the walk test using a standardised pain scale for claudication.[41] The walk test will be completed twice, with the highest 6MWD used for analysis.

### Secondary outcome measures
Other measures will be used to assess the broader effects of the MOSAIC intervention.

**Table 3** MOSAIC trial summary of measures

| Measure | Administered by | Baseline | 3 months | 6 months |
|---|---|---|---|---|
| **Baseline measures** | | | | |
| Sociodemographic characteristics | Self | | | |
| San Diego Claudication Questionnaire | Self | | | |
| Ankle-Brachial Pressure Index | Researcher | | | |
| Body mass index | Researcher | | | |
| **Primary outcome** | | | | |
| 6 min walking distance | Researcher | X | X | |
| **Secondary outcomes** | | | | |
| Pain-free walking time | Researcher | X | X | |
| Maximal walking time | Researcher | X | X | |
| Self-reported maximal walking distance* | Self | X | X | X |
| Walking Estimated-Limitation calculated by History | Self | X | X | X |
| Nottingham Extended Activities of Daily Living scale | Self | X | X | X |
| Vascular Quality of Life-6* | Self | X | X | X |
| EQ-5D-5L | Self | X | X | X |
| Brief International Physical Activity Questionnaire* | Self | X | X | X |
| Client Services Receipt Inventory | Self | X | X | X |
| **Process measures** | | | | |
| Brief Illness Perceptions Questionnaire | Self | X | X | X |
| Theory of Planned Behaviour Questionnaire | Self | X | X | X |
| Action Planning and Action Control | Self | X | X | X |
| Exercise Adherence Rating Scale | Self | | X | X |
| Adverse events* | Self/physiotherapist/ researcher | | X | X |

*Minimum dataset.
MOSAIC, MOtivating Structured walking Activity in people with Intermittent Claudication.

Maximal walking ability (duration walked before resting (seconds)) and pain-free walking ability (duration walked before reported pain onset (seconds)) will be recorded during the 6 min walk test.[42 43] These parameters evaluate the global and integrated physiological responses of all systems involved with exercise and symptom manifestation.

Self-reported maximal walking distance (SR-MWD) will be measured by one global item: 'What is the maximum distance (in metres) you can walk at your usual pace on a flat surface before leg pain forces you to stop?'[44] and the 4-item Walking Estimated-Limitation Calculated by History questionnaire (WELCH).[45 46]

Nottingham Extended Activities of Daily Living scale (NEADL)[47 48] is a 22-item measure with 4 subscales (mobility, kitchen tasks, domestic tasks and leisure activities) and will assess function.

The 6-item disease specific Vascular Quality of Life Questionnaire-6 (VascuQol-6)[49] and the EQ-5D-5L will assess QoL.[50]

The 7-item Brief IPAQ will estimate daily physical activity.[35]

Adverse events will be recorded by asking participants a single open-ended item: 'Have you had any problems since your last assessment?' at 3 and 6-month follow-up. Trial physiotherapists will also report any adverse events in participants randomised to receive the MOSAIC intervention.

To determine the MCID, participants will provide a global rating of change score after five measures (6MWD, SR-MWD, WELCH, NEADL and VascuQoL-6) at the 3 and 6-month follow-up assessments. In response to the question: 'Has there been any change in your walking ability/ walking distance/daily activities etc since the last test?' participants will be asked to rate their perceived change on a transitional three-point scale (1, worse; 2, about the same; 3, better). If they indicate no change, the participant will score 0. If they indicate there has been an improvement or deterioration, they will be asked to score their change on a 15-point Likert scale: (−7=a very great deal worse to 7=a very great deal better). Scores of −1, 0 and 1 will be considered no change, scores of 2–3 small improvement and scores of 4–7 substantial improvement.[51 52]

### Process variables

Process variables will be collected to investigate the implementation and maintenance, unexpected pathways and consequences, and mechanisms of impact of MOSAIC.

Attendance at MOSAIC sessions will be recorded by the trial physiotherapists. Adherence to walking goals will be assessed by the 6-item Exercise Adherence Rating Scale[53 54] at 3 and 6-month follow-up.

Fidelity to the MOSAIC intervention will be assessed in a random subset of ≥10% of the audio-recorded MOSAIC sessions/physiotherapist by two independent raters using bespoke, standardised checklists.

The type and duration of usual care and non-NHS care received by both groups will be recorded using the modified Client Service Receipt Inventory[36] to evaluate unexpected pathways and consequences.

Proposed mechanisms of impact include changes to theoretically defined sociocognitive variables targeted by MOSAIC. The Brief Illness Perception Questionnaire (a 9-item measure of individuals' representation of their illness as defined by the common-sense model),[55] the Theory of Planned Behaviour Questionnaire (a 12-item measure of goals and beliefs about walking as treatment for intermittent claudication)[32] and the action planning and action control scale (a self-regulation questionnaire)[56 57] will be assessed at each time point.

### Resource use variables

The study is an important opportunity to assess the feasibility of collecting data on resource use in adults with intermittent claudication to inform future trials. A modified Client Service Receipt Inventory[36] will be used to record information on service utilisation, income, accommodation and other cost-related variables at all time points to determine completion rates and redundant questions.

### Qualitative study

To explore the experience of MOSAIC treatment and participation in the trial, semistructured, audio-recorded interviews will be conducted with up to 30 participants or until data saturation of themes is reached. A purposive sampling strategy will ensure that a range of participants are selected (eg, age, symptoms, group allocation and recruitment site). Interviews will be conducted via telephone or face to face (depending on participant preference) after completion of the final assessment. A subsample of physiotherapists will be interviewed to explore their experiences of training and delivering MOSAIC by an independent researcher, supported by experienced qualitative researchers in the trial team.[12 13 58 59] Interview schedules will be developed iteratively, and the questions asked may develop as insights from ongoing interviews and analyses reveal additional areas of relevance. Interviews will be transcribed verbatim, anonymised and analysed thematically by one researcher.[60] The initial codes will be cross-referenced with a second researcher and the development of themes will be discussed with the trial team to provide different perspectives on coding. Themes will be presented to a sample of interviewees to ensure resonance and plausibility of the themes.

### PATIENT AND PUBLIC INVOLVEMENT

User involvement is central to the design and management of the MOSAIC trial. Patients were involved in the identification of the research question and two patient advisers reviewed the protocol and assisted in the development of all patient-related materials. One patient adviser is a funding coapplicant. Our patient advisers will be members of the trial management group, contribute to the trial physiotherapist training and dissemination. All participants will be invited to request study results, if interested.

## DATA AND STATISTICAL ANALYSIS PLAN

### Sample size

Based on previous work,[61] 192 participants will be required to detect a mean 6MWD difference of 58 m (SD=111; α=0.05, 1-β=0.90), accounting for 20% attrition at 3-month follow-up.

### Statistical analysis

A statistical analysis plan will be written by the trial statistician (SE) and agreed with the trial management group and joint trial steering committee/data monitoring and ethics committee. Analyses of primary and secondary outcomes and process variables will be conducted on an intention-to-treat basis and all included participants will be analysed as randomised.

The primary outcome will be analysed using multiple regression and adjusted for baseline 6MWD and site. Results will be reported as the difference in mean 6MWD between the intervention and control groups with 95% CIs. Other continuous outcomes and process variables will be similarly analysed. If appropriate, a mediation analysis using linear regression will examine mechanisms of impact.

Descriptive statistics for sociodemographic and clinical characteristics, attendance at MOSAIC sessions, adherence to treatment and referral to or uptake of other treatments will be computed. Adverse events and serious adverse events will be described.

To determine the MCID, change scores for the five clinical measures will be calculated by subtracting the baseline result from the follow-up results for each participant. Correlations will be computed for participant self-assessment of performance scores and change in clinical outcomes. The mean change in scores for participants reporting no change, small improvement and substantial improvement will be compared by analysis of variance. The sensitivity and specificity for change in score to distinguish participants classified as changed (≥2) from those whose performance was unchanged (−1 to +1) will be calculated and a receiver operating characteristic curve obtained.[62] The data point corresponding to the upper left corner of the curve will represent the MCID.

As a sensitivity analysis, the MCID will be also calculated using a distribution-based approach. The SE of measurement for all participants scores will be used to estimate the MCID based on the following equation: $\sigma_1\sqrt{(1-r)}$ where $\sigma_1$ is the baseline SD and $r$ represents the intraclass correlation coefficient, which is a measure of the test–retest reliability of the scale. The intraclass correlation coefficient will be calculated using the baseline and follow-up scores for each participant. Using this method, 1 SE of measurement represents the estimated MCID.[63]

If the proportion of missing data is above 10%, multiple imputation will be considered as a sensitivity analysis for the primary outcome, and for the secondary outcomes used in the 3 and 6-month MCID.[64]

## DATA MANAGEMENT AND MONITORING

Data will be collected and retained in accordance with the Data Protection Act 1998, the General Data Protection Regulation 2016/679, and the Data Protection Policy of King's College London. Data for each participant will be identified by a unique identification number and entered onto a Food and Drug Administration compliant database system. Data will be stored separately from personal data to maintain confidentiality. The chief investigator will be the custodian of the data and the data will only be used by the trial team.

The trial management group will be responsible for the management of the MOSAIC trial and will be led by the chief investigator. It will comprise trial investigators, a patient advisor and a vascular surgeon who will meet regularly to discuss trial progress.

A trial steering committee/data monitoring and ethics committee will provide study oversight. It will be independently chaired and will include patient and public involvement representatives, independent, experienced physiotherapists, a psychologist and a statistician along with the chief investigator. It will be independent of both the trial team and sponsors and operate under an agreed charter (MOSAIC Trial Steering Committee/Data Monitoring and Ethics Committee Charter V.1.0 25 July 17).

## ETHICAL CONSIDERATIONS AND DISSEMINATION

Approvals were provided from the Research and Development departments at all participating sites. The chief investigator will submit and, if required, obtain approval from relevant parties for all substantial amendments to the original approved documents.

The trial will be completed in accordance with the Declaration of Helsinki, the International Conference for Harmonisation of Good Clinical Practice guidelines and the Research Governance Framework for Health and Social Care.

King's College London and Guy's and St Thomas' NHS Foundation Trust (R&D@gstt.nhs.uk) are the trial sponsors and the lead site (Guy's and St Thomas' NHS Foundation Trust) will monitor and audit this project to ensure compliance with the necessary legislation.

This is a low risk trial because MOSAIC is a non-invasive treatment which will be delivered by trained, registered physiotherapists. Any adverse events will be recorded by the trained, trial physiotherapists and reported to the trial team. Participants will also be offered the opportunity to report any adverse events at follow-up assessments. Any serious adverse events will be referred to the chief investigator immediately.

### Dissemination

While the trial is in progress, we will disseminate trial updates via a dedicated website. (https://www.kcl.ac.uk/lsm/research/divisions/hscr/research/groups/Rehabilitation/MOSAIC-Trial.aspx) and social media (http://blogs.kcl.ac.uk/mosaic/). At the end of the trial, findings

will be disseminated via patient forums, conferences presentations and peer-reviewed publications.

## CONCLUSIONS

The MOSAIC trial investigates the efficacy of a physiotherapist-led, theoretically informed, behavioural change intervention on walking ability compared with usual care in older adults with intermittent claudication. It includes objective and self-reported measures of walking ability but only follows participants for 6 months. The MOSAIC intervention addresses a gap in the recommended care pathway for management of intermittent claudication, where walking exercise is a first-line treatment. Future studies should investigate the effectiveness and cost effectiveness of MOSAIC.

**Author affiliations**

[1]Department of Population Health Sciences, King's College London, London, UK
[2]Department of Applied Health Research, University College London, London, UK
[3]Academic Department of Vascular Surgery, King's College London, London, UK
[4]Department of Vascular Surgery, Guy's and St Thomas NHS Foundation Trust, London, UK
[5]Institute of Pharmaceutical Sciences, Kings College London, London, UK

**Acknowledgements** Thank you to the patient and public involvement representatives who provided advice and feedback on the development of MOSAIC and conduct of the trial, Pip Mason, from Pip Mason Consultancy, who contributed to the delivery of the trial physiotherapists training and St George's Hospital Charity for supporting the trial physiotherapists at St George's Hospital NHS Foundation Trust. MGH was (in part) supported by the National Institute for Health Research (NIHR) Collaboration for Leadership in Applied Health Research and Care (CLAHRC) North Thames at Bart's Health NHS Trust. The views expressed are those of the authors and not necessarily those of the NHS, the NIHR or the Department of Health and Social Care.

**Contributors** LB led the study design and funding application and, as chief investigator, has oversight for the trial. MGH, JW, JLP, BM, GF and CS contributed to the trial design and were coapplicants for funding. MGH, LB and JW developed the MOSAIC protocols and therapist training. JB is the trial coordinator and outcome assessor. LB, BV and JW trained and supervised the trial physiotherapists delivering MOSAIC. JLP led the statistical analysis and SE is the trial statistician. GF is an expert patient advisor and SP is leading recruitment and trial management at Guys and St Thomas' NHS Foundation Trust. All authors read and approved the final manuscript.

**Funding** This work was supported by The Dunhill Medical Trust (grant number R477/0516).

**Competing interests** None declared.

**Patient consent for publication** Not required.

**Ethics approval** The trial protocol was approved on 27 April 2017 by the London—Bloomsbury Research Ethics Committee (17/LO/0568). The current protocol is version 6 (approved on 28th August 2018).

**Provenance and peer review** Not commissioned; externally peer reviewed.

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
