## [Reviewer comments · BMJ Open]

ARTICLE DETAILS

TITLE (PROVISIONAL)	MOtivating Structured walking Activity in people with Intermittent Claudication (MOSAIC): protocol for a randomised controlled trial of a physiotherapist-led, behaviour-change intervention versus usual care in adults with intermittent claudication.
AUTHORS	Bearne, Lindsay; Galea Holmes, Melissa; Bieles, Julie; Eddy, Saskia; Fisher, Graham; Modarai, Bijan; Patel, Sanjay; Peacock, Janet; Sackley, Catherine; Volkmer, Britannia; Weinman, John

VERSION 1 – REVIEW

REVIEWER	Garry Tew Northumbria University, UK
REVIEW RETURNED	01-Mar-2019

GENERAL COMMENTS	I am pleased that the DMT funded the MOSAIC trial, and I eagerly await the outcomes. I hope my comments are helpful in improving the clarity of this protocol paper. Keywords - I question the relevance of "internal medicine" and "vascular surgery". Please consider replacing these terms with ones that relate to PAD and physical activity. I think that you should have submitted a completed SPIRIT checklist - please check this requirement. It might prove useful to review your against this checklist (if you haven't already). Abstract and elsewhere - remove capital letters from "Peripheral Arterial Disease" Abstract, line 11 - add "ability" after "walking" Abstract, line 26 - "group allocation" instead of "participant allocation"? P4, line 11 - consider specifying what usual NHS treatment is so the reader understands the comparison P4, lines 32-33 - are undulating terrain and lack of clarity barriers that are unique to IC? I was expecting a better justification for the study. Is this study needed? Does it address an important unknown? Please improve the justification for the study. You may want to refer to the recent systematic review of Golledge et al. in BJS. P5, line 28 - how exactly do you assess unstable IC?
---

	Tables 1 and 2 appear to have been presented in the wrong order Consider adding n-values to the flow diagram and tidying the positioning of the arrows P7 - the table specifies some of the behaviour change techniques, but it might be useful to list the included BCTs in the main body of the text. I also think it's very important to add what the goal of MOSAIC is. Is it just to walk a little bit more? Are there specific targets for frequency, intensity, duration etc...? What about progression? P8 - in the description of usual care, the term "pharmacological therapy" is too vague. For example, this could mean peripheral vasodilators or drugs to manage CV risk factors (e.g. hypertension, diabetes). Please be more specific here. Also, do both arms receive usual care? This isn't clear from the current description. Do any of the recruiting centres offer supervised or structured exercise programmes routinely? P10 - Pain-free and maximum walking times are usually assessed with a treadmill protocol where the speed is controlled, and the same at baseline and follow-up. Here, a self-paced test is used. Please justify your approach and comment on the potential impact of differing walking speeds between baseline and follow-up. I think you also need to be clearer that 6MWD is the primary outcome measure because several outcome measures are listed in the primary outcome section. P13 - for the sample size calculation, please can you clarify where the MD of 58m and SD of 111m come from? I can't see these values in the GOALS protocol paper which you've cited. Please can you also clarify why you used their protocol paper instead of their results? Looking at their 2013 results paper in JAMA, the SD of 111m seems reasonable, but is a MD of 58m unrealistic? They observed a MD of 53.5m at 6 months after a more intensive and prolonged walking intervention. How will missing data be handled?
--	---

REVIEWER	Ukachukwu Abaraogu Glasgow Caledonian University United Kingdom
REVIEW RETURNED	04-Apr-2019

GENERAL COMMENTS	This is an excellent and well thought out research protocol and when conducted will be a timely contribution to literature on PAD and self-management strategies. Readers will benefit from a brief discussion section and possibly highlighting/acknowledging known limitations in the protocol just before the conclusion.
--

REVIEWER	Nicola Lamberti Department of Biomedical and Surgical Specialties Sciences, University of Ferrara, Italy
REVIEW RETURNED	23-Apr-2019

GENERAL COMMENTS	Authors present a study protocol dealing with a new behavioral-change intervention in PAD. I congratulate the authors for their trial, and I will be waiting for their important results. I have only some minor issues to be addressed. 1) Introduction: Please use PAD acronym throughout the text (maybe as peripheral artery disease, MeSH term) Please expand the NHS word Please mention some structured home-based programs as promising tools for PAD patients (opinion from last AHA statement) 2) Methods Please specify if a blinded assessor will be present in each center, or the same assessor will travel to each center Please specify if any change in PAD-specific medication will be allowed during the study (e.g. the use of Cilostazol, which can sensibly modify your outcomes) Please do not use pain-free and resting walking time but as usually performed, for the 6MWT use the 6-minute walking distance, the distance covered at the first stop, and the pain-free walking distance. Walking time actually is enormously influenced by the speed. For example: a patient that walk at a 100 steps/min can present claudication after 100 meters, or 2 minutes. If the same patients walks at 60 steps/min can present claudication after 100 meters, but 5 minutes. That is the major issue to be addressed. Please give more details about the walking program proposed to the patients, and especially details about how you will collect and measure adherence to the program (e.g. distance walked for each session) 3) Statistical analysis The difference of 58 meters is in favor of which group? Why for primary outcomes (difference in change of 6MWD from baseline to end of the program between groups) did you not employ a standard independent samples t-test or a two-way repeated measures (factors treatment, time) ANOVA? Please mention in the manuscript that is written according to the SPIRIT guidelines and checklist (and provide reference if you can).
--

VERSION 1 – AUTHOR RESPONSE

Responses to Reviewer 1:

Keywords - I question the relevance of "internal medicine" and "vascular surgery". Please consider replacing these terms with ones that relate to PAD and physical activity.

Key words have been amended

I think that you should have submitted a completed SPIRIT checklist - please check this requirement. It might prove useful to review your against this checklist (if you haven't already).

The SPIRIT checklist and an example consent form has been submitted as supplementary documents

Abstract and elsewhere - remove capital letters from "Peripheral Arterial Disease"

This has been amended throughout to peripheral arterial disease or PAD, as appropriate

Abstract, line 11 - add "ability" after "walking"

This has been amended, as suggested

Abstract, line 26 - "group allocation" instead of "participant allocation"?

This has been amended, as suggested

P4, line 11 - consider specifying what usual NHS treatment is so the reader understands the comparison

Further details of recommended management of PAD have been included in the manuscript

P4, lines 32-33 - are undulating terrain and lack of clarity barriers that are unique to IC?

This sentence has been rephrased in the text to improve accuracy

I was expecting a better justification for the study. Is this study needed? Does it address an important unknown? Please improve the justification for the study. You may want to refer to the recent systematic review of Golledge et al. in BJS.

The introduction has been revised to include greater justification for the MOSAIC trial and cites recent systematic reviews

P5, line 28 - how exactly do you assess unstable IC?

Unstable IC is assessed by self-report in response to the question 'Have your symptoms changed in the last 3 months?'). This information has been added to the manuscript

Tables 1 and 2 appear to have been presented in the wrong order

The table order has been revised and an additional table added

Consider adding n-values to the flow diagram and tidying the positioning of the arrows

n-values have been added to the randomisation and group allocation in the trial flow chart. The diagram, including arrows, have been aligned and formatted

P7 - the table specifies some of the behaviour change techniques, but it might be useful to list the included BCTs in the main body of the text. I also think it's very important to add what the goal of MOSAIC is. Is it just to walk a little bit more? Are there specific targets for frequency, intensity, duration etc...? What about progression?

Additional details of the aims of the MOSAIC, walking progression and behaviour change techniques included in the intervention have been added to the main text and in an additional table

P8 - in the description of usual care, the term "pharmacological therapy" is too vague. For example, this could mean peripheral vasodilators or drugs to manage CV risk factors (e.g. hypertension, diabetes). Please be more specific here. Also, do both arms receive usual care? This isn't clear from the current description. Do any of the recruiting centres offer supervised or structured exercise programmes routinely?

Details of usual care have been included in the manuscript. Two centres offer a medically prescribed supervised exercise programme for some of their patients with intermittent claudication (depending on the healthcare commissioned). However, patients are not eligible to be enrolled into the MOSAIC trial if they are due to commence a medically prescribed supervised exercise programme for intermittent

claudication within the upcoming 6 months or until at least 6 months after completing a medically prescribed supervised exercise programme for intermittent claudication

P10 - Pain-free and maximum walking times (PFWT) are usually assessed with a treadmill protocol where the speed is controlled, and the same at baseline and follow-up. Here, a self-paced test is used. Please justify your approach and comment on the potential impact of differing walking speeds between baseline and follow-up. I think you also need to be clearer that the 6 Minute Walking Distance (6MWD) is the primary outcome measure because several outcome measures are listed in the primary outcome section.

The primary outcome for the MOSAIC trial is the difference in mean 6MWD at 3 months between the intervention and comparison groups assessed by a corridor based 6 minute walk test. This has been clarified in the manuscript.

We will be collecting PFWT and MWT as secondary outcome measures, as described previously (1). We agree that the PFWT and MWT assessed during a self-paced corridor 6MWT may be affected by a change in speed. However, in controlled circumstances, such as a standardized 6MWT, these parameters are highly replicable (2, 3). As we will be conducting two 6MWTs at each assessment we will examine the data from these two tests for consistency at each timepoint. However, we acknowledge that the interpretation of the PFTW and MWT may be different when conducted in a self-paced corridor 6MWT. Whilst these parameters evaluate the global and integrated physiological responses of all the systems involved during exercise in a standardized treadmill test, in a self-paced corridor test, we anticipate they will also capture a manifestation of a patient's symptoms. Data on these changes in symptomology in these tests will provide important and novel data to describe walking following a structured home-based walking programme

P13 - for the sample size calculation, please can you clarify where the MD of 58m and SD of 111m come from? I can't see these values in the GOALS protocol paper which you've cited. Please can you also clarify why you used their protocol paper instead of their results? Looking at their 2013 results paper in JAMA, the SD of 111m seems reasonable, but is a MD of 58m unrealistic? They observed a MD of 53.5m at 6 months after a more intensive and prolonged walking intervention.

Unfortunately, the incorrect reference was provided for the sample size calculation, this reference has been corrected in the revised version. In table 2 (4), six month follow-up is shown to be 399.8 (101.6) vs 342.2 (110.8), so a difference in means of 57.6m (399.8 – 342.2) or 58m (rounded to the nearest integer) is plausible

How will missing data be handled?

If the proportion of missing data is above 10% (5) multiple imputation may be considered as a sensitivity analysis for the primary outcome, and for the secondary outcomes used in the three and six month MCIDs. A complete data set analysis will be the main analysis

Responses to Reviewer: 2

This is an excellent and well thought out research protocol and when conducted will be a timely contribution to literature on PAD and self-management strategies. Readers will benefit from a brief discussion section and possibly highlighting/acknowledging known limitations in the protocol just before the conclusion.

Thank you for your comments - a brief discussion section has been included and study limitations highlighted in the strengths and limitation section of the manuscript

Responses to Reviewer: 3

Authors present a study protocol dealing with a new behavioral-change intervention in PAD. I congratulate the authors for their trial, and I will be waiting for their important results.

I have only some minor issues to be addressed.

Thank you for your comments

1) Introduction:

Please use PAD acronym throughout the text (maybe as peripheral artery disease, MeSH term).

Please expand the NHS word.

These amendments have been addressed throughout

Please mention some structured home-based programs as promising tools for PAD patients (opinion from last AHA statement)

This has been included in the introduction

2) Methods

Please specify if a blinded assessor will be present in each center, or the same assessor will travel to each center

This has been clarified in the manuscript

Please specify if any change in PAD-specific medication will be allowed during the study (e.g. the use of Cilostazol, which can sensibly modify your outcomes)

There are no limitations on the healthcare prescribed by the direct care team in either group. Any changes in healthcare will be recorded on a modified Client Receipt Service Inventory at each follow up assessment. This has been clarified in the manuscript

Please do not use pain-free and resting walking time but as usually performed, for the 6MWT use the 6-minute walking distance, the distance covered at the first stop, and the pain-free walking distance. Walking time actually is enormously influenced by the speed. For example: a patient that walk at a 100 steps/min can present claudication after 100 meters, or 2 minutes. If the same patients walks at 60 steps/min can present claudication after 100 meters, but 5 minutes. That is the major issue to be addressed.

We will be collecting PFWT and MWT as secondary outcome measures, as described previously (1). These measures have been approved by the ethical review committee and supported by the trial steering/data monitoring and ethics committee. We agree that the PFWT and MWT assessed during a self-paced corridor 6MWT may be affected by a change in speed. However, in controlled circumstances, such as a standardized 6MWT, these parameters are highly replicable (2, 3). As we will be conducting two 6MWTs at each assessment we will examine the data from these two tests for consistency at each timepoint. However, we acknowledge that the interpretation of the PFWT and MWT may be different when conducted in a self-paced corridor 6MWT. Whilst these parameters evaluate the global and integrated physiological responses of all the systems involved during exercise in a standardized treadmill test, in a self-paced corridor test, we anticipate they will also capture manifestations of patient's symptoms. Data on these changes in symptomology in these tests will provide important and novel data to describe walking following a structured home-based walking programme

Please give more details about the walking program proposed to the patients, and especially details about how you will collect and measure adherence to the program (e.g. distance walked for each session).

Further details of the MOSAIC and the behaviour change techniques included in the intervention have been incorporated in the manuscript. Attendance at MOSAIC sessions will be recorded by the trial

physiotherapists and adherence to walking goals will be assessed at follow-up by the 6-item Exercise Adherence Rating Scale

3) Statistical analysis

The difference of 58 meters is in favor of which group?

Why for primary outcomes (difference in change of 6MWD from baseline to end of the program between groups) did you not employ a standard independent samples t-test or a two-way repeated measures (factors treatment, time) ANOVA?

Thank you for your comment. Perhaps this does not come across clearly in the protocol, but, the primary outcome is: the difference in mean 6MWD at 3 months between the intervention and comparison groups, adjusted for baseline 6MWD and site. A t-test cannot be used, as we need to adjust for differences in baseline values and group differences in site. In addition, analysing change does not adjust for baseline imbalances due to regression to the mean, hence why ANCOVA/regression models are typically utilised in the analysis of clinical trials (6).

References

1. Dixit S, Chakravarthy K, Reddy RS, Tedla JS. Comparison of two walk tests in determining the claudication distance in patients suffering from peripheral arterial occlusive disease. *Adv Biomed Res.* 42015.
2. McDermott MM, Guralnik JM, Criqui MH, Liu K, Kibbe M, Ferrucci L. The Six-Minute Walk is a Better Outcome Measure than Treadmill Walking Tests in Therapeutic Trials of Patients with Peripheral Artery Disease. *Circulation.* 2014;130(1):61-8.
3. Galea Holmes M.N. Increasing walking in individuals with intermittent claudication: the roles of walking treatment and illness cognitions. King's College London 2016.
4. McDermott M.M, Liu K, Guralnik J.M, Criqui M.H, Spring B, Tian L, et al. Home-Based Walking Exercise Intervention in Peripheral Artery Disease: A Randomized Clinical Trial. *JAMA.* 2013;310(1):57-65.
5. Bennett DA. How can I deal with missing data in my study? *Australian and New Zealand Journal of Public Health.* 2001;25(5):464-9.
6. Vickers AJ, Altman DG. Analysing controlled trials with baseline and follow up measurements. *BMJ.* 2001;323(7321):1123-4.

VERSION 2 – REVIEW

REVIEWER	Garry Tew Northumbria University, United Kingdom
REVIEW RETURNED	11-Jun-2019

GENERAL COMMENTS	Thank you for responding the the reviewers' comments and revising the manuscript. The responses and edits are satisfactory, however I cannot see the SPIRIT checklist and example consent form in the review portal, which you said were submitted as supplementary documents. I also have one new minor comment: Where you describe the primary outcome, I think that it would be helpful to be more specific when you say that the best test will be used for analysis; so what exactly is meant by "best"? I presume you mean the highest 6MWD, but I think it's worth you clarifying this point.
--

REVIEWER	Nicola Lamberti University of Ferrara, Italy
REVIEW RETURNED	05-Jun-2019

GENERAL COMMENTS	Authors have successfully address all my comments.
--

VERSION 2 – AUTHOR RESPONSE

Reviewer(s)' Comments to Author:

Reviewer: 1

Thank you for responding the reviewers' comments and revising the manuscript. The responses and edits are satisfactory, however I cannot see the SPIRIT checklist and example consent form in the review portal, which you said were submitted as supplementary documents.

Thank you for your comments. We have uploaded the SPIRIT checklist and consent form again as supplementary files for consideration by the reviewers.

I also have one new minor comment: Where you describe the primary outcome, I think that it would be helpful to be more specific when you say that the best test will be used for analysis; so what exactly is meant by "best"? I presume you mean the highest 6MWD, but I think it's worth you clarifying this point.

Thank you - we have amended the text for clarity.

Reviewer: 3

Authors have successfully addressed all my comments.

Thank you

VERSION 3 – REVIEW

REVIEWER	Garry Tew Northumbria University
REVIEW RETURNED	19-Jun-2019

GENERAL COMMENTS	The reviewer completed the checklist but made no further comments.
--